# Preferential Localization of MUC1 Glycoprotein in Exosomes Secreted by Non-Small Cell Lung Carcinoma Cells

**DOI:** 10.3390/ijms20020323

**Published:** 2019-01-14

**Authors:** Deng Pan, Jiaxi Chen, Chunchao Feng, Weibo Wu, Yanjin Wang, Jiao Tong, Dapeng Zhou

**Affiliations:** Tongji University School of Medicine, Shanghai 200092, China; dengpan88@163.com (D.P.); chenjiaxi753@tongji.edu.cn (J.C.); chunchao0825@163.com (C.F.); violetwwb@163.com (W.W.); wangyanjin0317@163.com (Y.W.); tongjiao2109@163.com (J.T.)

**Keywords:** exosome, non-small cell lung cancer, MUC1, biomarker, exosomal proteomics

## Abstract

Lung cancer remains to be the leading cause of cancer-related mortality worldwide. Finding new noninvasive biomarkers for lung cancer is still a significant clinical challenge. Exosomes are membrane-bound, nano-sized vesicles that are released by various living cells. Studies on exosomal proteomics may provide clues for developing clinical assays. In this study, we performed semi-quantitative proteomic analysis of proteins that were purified from exosomes of NCI-H838 non-small cell lung cancer cell line, with total cellular membrane proteins as control. In the exosomes, LC-MS/MS by data-independent analysis mode identified 3235 proteins. THBS1, ANXA6, HIST1H4A, COL18A1, MDK, SRGN, ENO1, TUBA4A, SLC3A2, GPI, MIF, MUC1, TALDO1, SLC7A5, ICAM1, HSP90AA1, G6PD, and LRP1 were found to be expressed in exosomes at more than 5-fold higher level as compared to total cellular membrane proteins. A well-known cancer biomarker, MUC1, is expressed at 8.98-fold higher in exosomes than total cellular membrane proteins. Subsequent analysis of plasma exosomes from non-small cell lung cancer (NSCLC) patients by a commercial electrochemiluminescence immunoassay showed that exosomal MUC1 level is 1.5-fold higher than healthy individuals (mean value 1.55 ± 0.16 versus mean value 1.05 ± 0.06, *p* = 0.0213). In contrast, no significant difference of MUC1 level was found between NSCLC patients and healthy individuals′ plasma (mean value 5.48 ± 0.65 versus mean value 4.16 ± 0.49). These results suggest that certain proteins, such as MUC1, are selectively enriched in the exosome compartment. The mechanisms for their preferential localization and their biological roles remain to be studied.

## 1. Introduction

Lung cancer is one of the most fatal malignancies and the leading cause of tumor mortality in both men and women in the world, with the incidence of 1.8 million new cases and 1.6 million death per year [1,2]. There are two types of lung cancer, non-small cell lung cancer (NSCLC) accounts for about 85% of lung cancers, and small cell lung cancer (SCLC) accounts for 15% of lung cancers [3]. Surgery is the recommended treatment for patients with stage I–II diseases [4,5]. Five-year survival is 77–92%, 68%, 60%, and 53% for clinical stage IA, IB, IIA, and IIB, respectively [6,7]. However, in all NSCLC cases, approximately 75% cases were diagnosed at advanced stage of disease, and the five-year overall survival rate is only 15% for all stages’ patients in spite of recent development of targeted therapy and immune blockade antibodies [8,9]. Traditional diagnosis of NSCLC tends to be based on X-ray, computerized tomography (CT), and pneumonocentesis [10]. Unfortunately, it has been estimated that less than 30% of cases can be detected at an early stage when the curative surgery is possible [11]. Moreover, the potential over-diagnosis and harmful effects induced by radiation limit their clinical use [12]. Consequently, novel and reliable biomarkers to detect NSCLC for early intervention with the potential to reduce mortality are urgently needed.

Exosomes are small membrane vesicles, with a density of 1.13–1.21 g/mL, and a diameter of 30–150 nm [13,14]. Exosomes are secreted by most cell types, including cancer cells. The contents of exosomes contain nucleic acids (DNA, mRNA, and microRNA), proteins, and lipids [15]. Proteins enriched in exosomes include tetraspanin family members (CD9, CD63, CD81, and CD82), members of the endosomal sorting complexes that are required for transport (Alix and TSG101) and heat-shock proteins (Hsp60, Hsp70, and Hsp90) [16]. Exosomes can be purified from several body fluids, such as serum, plasma, urine, salivary, and cerebrospinal fluid. Several studies found that the concentration of exosomes in the plasma of patients with tumor is higher than healthy control [17,18]. As a diagnostic biomarker, exosomes have been evaluated in several tumors, such as breast cancer, ovarian cancer, and glioblastoma [17,19,20]. In lung cancer, several studies have found the diagnostic perspectives of exosomal miRNA and proteins in the plasma [21,22,23]. Exosomal protein, such as EGFR, has been proposed as a possible marker for lung cancer diagnosis.

In this study, we hypothesized that certain glycoproteins are selectively enriched in exosomes, and performed proteomics analysis of exosomal proteins and total cellular membrane proteins of NCI-H838, a non-small cell lung carcinoma cell line. A well-known biomarker, MUC1 was identified as a most abundant glycoprotein with preferential expression in exosome.

## 2. Results

### 2.1. Characterization of Exosomes

CD63, one of the most commonly used for the identification of exosome biomarker, was measured by Western blot for purified exosomal proteins (Figure 1A). Nanoparticle tracking analysis (NTA) was used to evaluate the size distribution of exosomes. Exosomes that were extracted by ultracentrifugation or exoEasy Maxi Kit were detected as a homogenous population with low dispersity by NTA. Exosomes extracted by two methods showed as a peak with a particle size of approximately 145 nm and 165 nm, respectively (Figure 1B,C).

### 2.2. Proteins Preferentially Localized in the Exosomes of NCI-H838 Cells

We performed semi-quantitative proteomic analysis of proteins that were purified from exosomes of NCI-H838 non-small cell lung cancer cell line, with total cellular membrane proteins as control. LC-MS/MS by data-independent analysis mode identified 3235 proteins in exosomes versus 4363 proteins in total cellular membrane proteins (Appendix A). 2929 proteins were found both in exosomes and membrane of NCI-H838 cells, among which 42 proteins were reported by several groups as potential biomarkers for lung cancer (Figure 2, Table 1 [23,24,25,26,27,28,29,30,31,32,33,34,35,36,37,38,39,40]).

We next searched for proteins that are preferentially expressed in exosomes. 685 proteins were found to be selectively enriched in exosomes at more than five-fold higher level as compared to total cellular membrane proteins (Appendix A), including 18 proteins, which were previously reported as potential lung cancer biomarkers, THBS1, ANXA6, HIST1H4A, COL18A1, MDK, SRGN, ENO1, TUBA4A, SLC3A2, GPI, MIF, MUC1, TALDO1, SLC7A5, ICAM1, HSP90AA1, G6PD, and LRP1 (Table 1). Interestingly, several well-known cancer biomarkers, THBS1, ENO1, and MUC1, were found to be expressed at much higher abundance in exosomes than total membrane proteins. When compared to total cellular membrane fraction of NCI-H838 cells, THBS1, ENO1, and MUC1 in exosomes increased 259.86, 16.10, and 8.98-fold, respectively (Table 1). The presence of MUC1 in the exosomes of NCI-H838 cells was further confirmed by western blot (Figure 3). Furthermore, the presence of MUC1 in the exosomes of NCI-H838 cells was confirmed by mapping for trypsin-digested peptide fragments (Appendix A).

### 2.3. Exosomal Expressed MUC1 Is Significantly Different between NSCLC Patients and Healthy Controls

Since MUC1 is a widely used biomarker in cancer clinics, we tried to measure the exosomal MUC1 by commercially available tests that are currently used in clinics. We used the exoEasy Maxi Kit from Qiagen to enrich exosome fractions. The difference in levels of MUC1 in the NSCLC patients and healthy controls was detected by electrochemiluminescence immunoassay analysis (EIA). When compared with the healthy controls, the levels of MUC1 in the NSCLC patients’ enriched exosomes was up-regulated, and the difference was statistically significant (mean value 1.55 ± 0.16 versus mean value 1.05 ± 0.06, *p* = 0.0213) (Figure 4A, Table 2). However, no significant difference was found in plasma of NSCLC patients and healthy controls (mean value 5.48 ± 0.65 versus mean value 4.16 ± 0.49) (Figure 4B, Table 2). Moreover, we constructed a receiver operating characteristic (ROC) curve to evaluate the diagnostic value of exosomal MUC1 for NSCLC. The area under the curve (AUC) for plasma exosomal MUC1 was 0.685 (95% CI: 0.526–0.818, *p* = 0.0234) (Figure 4C). However, the AUC for plasma MUC1 was 0.569 (95% CI: 0.410–0.719, *p* = 0.4463). These results suggested that exosomal MUC1 might be valuable in distinguishing NSCLC patients from healthy controls.

## 3. Discussion

In recent years, the role of exosomes in tumor diagnosis is of particular interest because tumor cells secrete exosomes at least 10-fold more than normal cells [42]. Vykoukal et al. reported the proteomics analysis of plasma-derived exosomes in lung adenocarcinoma patients (13 cases versus 15 controls). They reported that SRGN, TPM3, THBS1, HUWE1, and CCDC18 were enriched in lung adenocarcinoma extracellular vesicles. When compared to healthy controls, SRGN, TPM3, THBS1, HUWE1, and CCDC18 increased 5.30, 44.1, 5.27, 8.36, and 9.77-fold, respectively [26]. However, MUC1 was not identified in Vykoukal′s study, which might be due to the difficulty in sample preparation to generate sufficient MUC1 peptides from plasma exsomes. Technically, we used LTQ Orbitrap Fusion Lumosmass spectrometer, which identified 3525 exosomal proteins from lung cancer cells. While Vykoukal et al. used LTQ-Orbitrap XL mass spectrometer, which identified 680 exosomal proteins from plasma of lung cancer patients.

Sandfeld-Paulsen et al. used 49 antibodies microarray, including an anti-MUC1 antibody, to capture MUC1 on the exosomes of NSCLC patients’ plasma, and the MUC1 positive detection rate is up to 80% [43]. This indicates that plasma exosomal MUC1 has potential as a diagnostic marker for NSCLC. In another study, Sun et al. reported that MUC1 could be identified from salivary extracellular vesicles among 12 candidate proteins that were identified by comparative proteomics [33]. In the present study, due to limitations in preparing sufficient exosome-derived peptides from patient plasma, we chose to use commercially available electrochemiluminescence immunoassay to measure the exosomal MUC1. We found a two antibodies combination for diagnosis with high sensitivity and specificity for the detection of plasma exosomal MUC1 in NSCLC patients. We report here that the exosomal MUC1, but not plasma MUC1, is valuable in distinguishing NSCLC and healthy controls.

MUC1 is highly polarized to the apical surface of most epithelial cells. Yet, during tumorigenesis, MUC1 is depolarized. In normal tissues, MUC1 is highly *O*-glycosylated, with 50–90% of the protein weight being carbohydrate [44]. In tumor cells, the glycoprotein MUC1 undergoes aberrant, truncated glycosylation. Previous researchers have reported the most common glycan structures in lung cancer, including Tn antigen (GalNAc), STn antigen (Neu5Acα2-6GalNAc), and ST antigen (NeuAcα1-3Gaβ1-3GalNAc) [44,45]. In a normal airway, the expression of MUC1 is weak and MUC1 expression is not found in the bronchioles [46]. However, several reports suggested that MUC1 was overexpressed in NSCLC and can be used as a suitable biomarker for diagnosis of lung cancer [47,48]. MUC1 was significantly increased in adenocarcinoma (ADC) but not in squamous cell carcinoma (SCC) through glycoproteomics and proteomic analysis of NSCLC tissue samples [25,49]. Moreover, Ring et al. showed that a five-antibody (include MUC1 antibody) test is able to reproducibly distinguish ADC and SCC tumors [32]. These results indicated that MUC1 may be specific to ADC subtype. However, several studies through proteomic analysis of plasma samples failed to detect MUC1 [24,26,29]. Multiple proteins identified in those studies include CYFRA21-1, CEA, SCC, MMP2, CRP, NSE, CA125, ICAM1, ENO1, SRGN, TPM3, THBS1, HUWE1, and zyxin. The failure to detect MUC1, a most widely used cancer biomarker, might be due to technical difficulties to generate sufficient peptides by trypsin digestion.

It is now generally accepted that multiple protein markers are superior in early diagnosis of NSCLC. For example, CYFRA21-1, as the best single marker was able to only detect 23.3% of NSCLC at stage I [50]. Our findings of preferential localization of MUC1 in exosomes suggest that exosomal MUC1 may be valuable for the panel of protein biomarkers for NSCLC.

MUC1 has been proposed as a signaling molecule that is involved in regulating the apoptotic pathway through binding to BAX protein [51]. The glycans of MUC1 have been suggested as ligands for galectins and/or Siglecs expressed on the surface of lymphocytes [52,53], which trigger the apoptosis of tumor infiltration lymphocytes. Our preliminary findings suggest that the glycoforms of exosomal MUC1 is different from total cellular membrane proteins. The exact glycan structures of exosomal MUC1 will be future research focus, as well as their pathobiological functions in tumor microenvironment and tumor immunity.

## 4. Materials and Methods

### 4.1. Patients and Cell Line

Plasma samples from patients with NSCLC who had no surgery within past 12 months were received from Shanghai Pulmonary Hospital affiliated with Tongji University School of Medicine (Appendix A). The samples contain 27 NSCLC and 16 healthy controls. The experiment was approved by the Ethics Committee of Shanghai Pulmonary Hospital affiliated with Tongji University School of Medicine (No. K16-245-1, 29 Feb 2016). The human tumor cell line NCI-H838 was obtained from the America Type Culture Collection (ATCC) and was cultured at 37 °C with 5% CO_2_ in RPMI-1640 media (Life Technologies, Carlsbad, CA, USA), with 10% fetal bovine serum (Life Technologies, Carlsbad, CA, USA). Culture media was switched to serum-free medium before the harvesting of secreted exosomes.

### 4.2. Western Blot Analysis

Exosomal protein concentration was determined by the Bradford method and proteins were separated by SDS-PAGE gel electrophoresis. Proteins were transferred onto PVDF membranes (Millipore, Burlington, MA, USA). The blots were blocked for 2 h at room temperature with 5% non-fat milk in PBS/0.05% Tween 20 and then incubated with the primary antibody (1 µg/mL) overnight at 4 °C. After washing, the PVDF membrane was incubated with horseradish peroxidase (HRP)-conjugated secondary antibody at room temperature for 1 h. The transferred proteins were visualized with chemiluminescence detection kit (Millipore, Burlington, MA, USA).

### 4.3. Nanoparticle Tracking Analysis

Exosomes that were isolated from conditioned cell culture media and plasma were analyzed by nanoparticle tracking analysis, using the NanoSight LM10 system (NanoSight Ltd, Wiltshire, UK). Each sample was diluted 100-fold in PBS for counts in the linear range of the instrument. The particle movements are recorded and all of the samples were analyzed in triplicates.

### 4.4. Isolation of Exosomes

NCI-H838 cells were grown in 10 cm dishes with serum-free RPMI-1640 media for 72 h, and the cells reached a confluency of 90–100%. The media was collected and centrifuged at 3000× *g* at 4 °C for 10 min to remove detached cells. Subsequently, the media was centrifuged at 10,000× *g* at 4 °C for 30 min to remove large extracellular vesicles. Supernatant was collected and filtered through 0.22 µm filters (Millipore, Burlington, MA, USA) to remove microvesicles and contaminating apoptotic bodies. The supernatant was then centrifuged at 100,000× *g* in type 45 Ti fixed angle rotor using Optima L-80XP Ultracentrifuge (Beckman Coulter, Brea, CA, USA) at 4 °C for 2 h. The supernatant was carefully removed, and the pellets were resuspended in 70 mL of ice-cold PBS and then centrifuged at 100,000× *g* in type 45 Ti fixed angle rotor at 4 °C for 2 h. The PBS was carefully removed, and exosomes were resuspended in 200 µL of ice-cold PBS.

Green-top tube (containing sodium heparin) as an anticoagulant plasma separator tubes were used to collect blood samples. The blood samples were then centrifuged at 5000 rpm for 10 min to collect the plasma, which was stored at −80 °C until used. The plasma exosomes were extracted by exoEasy Maxi Kit (Qiagen, Venlo, The Netherlands) according to the manufacturer′s protocol. Briefly, plasma was centrifuged at 10,000× *g* at 4 °C for 30 min to remove large extracellular vesicles. Supernatant was collected and filtered through 0.22 µm filters (Millipore, Burlington, MA, USA) to remove microvesicles, contaminating apoptotic bodies. Afterwards, an equal volume of buffer XBP was added, mixed thoroughly, transferred into the exoEasy spin column, and centrifuged at 5000× *g* at 4 °C for 1 min. The flow-through was discarded and the spin column was washed by 10 mL buffer XWP. Then the spin column was transferred to a new collection tube, 200 µL buffer XE was added to elute the exosomes.

### 4.5. Mass Spectrometry Method and Data Analysis

An equal amount of proteins from each sample was digested using filter-aided sample preparation (FASP) [54]. A mixture sample was made of equal amount of digested peptides from each sample and then separated into three fractions using a modified High-pH reversed-phase method. For the generation of spectral library, the pre-separated three fractions were acquired with data dependent acquisition mode using Orbitrap Fusion Lumos (SanJose, Thermo Fisher, Waltham, MA, USA). Peptide mixture were separated on EasynLC LC1200 system (San Jose, Thermo Fisher, Waltham, MA, USA) using a home-made C18 column (3 µm, 75 µm × 15 cm) at a flow rate of 600 nL/min. For comparative proteomic analysis between two samples, peptide concentration from two samples was determined by NanoDrop 2000 and an equal amount of peptide from two samples with the addition of same amount of indexed retention time standards (iRT, Biognosys) was analyzed with DIA method. DDA raw files were searched against the uniprot protein database of Homo sapiens (9606) using Proteome Discoverer 2.1 (San Jose, Thermo Fisher, Waltham, MA, USA). The searching result of data-dependent acquisition using Proteome Discoverer 2.1 was transferred into a spectral library using Spectronaut 10 (Biognosys, Schlieren, Switzerland).

The following settings were applied in Spectronaut 10.0 peak detection, dynamic iRT, enabled, correction factor 1, dynamic score refinement and MS1 scoring, enabled, interference correction and cross run normalization (total peak area), enabled. The number of fragment ions was defined in the spectral library, and all were required for identification and quantification. Spectronaut utilizes the spiked-in iRT peptides for m/z and retention time calibration. For our data set, the m/z tolerance was in the range of 4 ppm and the median retention time extraction window 8 min. All of the results were filtered by a Q value of 0.01 (equals a FDR of 1% on peptide level). All other settings were set to default. Protein intensity was calculated by summing the peptide peak areas (sum of fragment ion peak areas, as calculated by Spectronaut) of each protein from the Spectronaut output file.

### 4.6. Electrochemiluminescence Immunoassay Analysis (EIA)

The plasma exosomes were extracted by exoEasy Maxi Kit (Qiagen, Venlo, The Netherlands). Plasma exosomal MUC1 concentration was detected by an EIA (elecsys immunoassay) system, developed by Tellgen (Shanghai, China), which uses electrochemiluminescence as detection technology, thereby offering the advantage of high sensitivity and short reaction time. Briefly, 25 µL sample was incubated with biotinylated M6809 capture antibody and ruthenylated M6801 tracer antibody at room temperature for 10 min. The sample was then incubated with streptavidin-coated magnetic beads for 10 min at room temperature. Subsequently, the reaction mixture was drawn into the detection cell, and the electrochemiluminescence was detected by a photomultiplier.

### 4.7. Statistical Analysis

The different MUC1 levels of exosomes were compared using unpaired student′s *t*-test in NSCLC and healthy control samples using GraphPad Prism version 5 (GraphPad software, La Jolla, CA, USA). A two-sided *p* value < 0.05 was defined as statistical significance.

## 5. Conclusions

This is the first report to find MUC1 in exosomes of lung cancer cells by LC-MS/MS. In this study, we performed a proteomics analysis of exosomal proteins and total cellular membrane proteins of NCI-H838, a non-small cell lung carcinoma cell line. A well-known cancer biomarker, MUC1, is expressed at nine-fold higher in exosomes than total membrane proteins. Subsequent analysis of plasma exosomes from NSCLC patients showed that the exsosomal MUC1 level is 1.5-fold higher than healthy individuals. These results suggested that MUC1 may be a sensitive biomarker that can distinguish NSCLC patients from healthy controls.

## Figures and Tables

**Figure 1 ijms-20-00323-f001:**
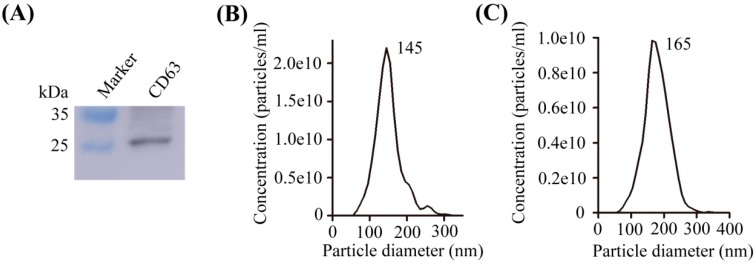
Characterization of exosomes isolated from NCI-H838 cells and patient plasma. (**A**) Western blotting analysis to measure expression of CD63 in exosome of NCI-H838 cells; (**B**) Nanoparticle tracking analysis of exosomes from NCI-H838 cells extracted by ultracentrifugation; (**C**) Nanoparticle tracking analysis of exosomes extracted from non-small cell lung cancer (NSCLC) patients’ plasma by exoEasy Maxi Kit.

**Figure 2 ijms-20-00323-f002:**
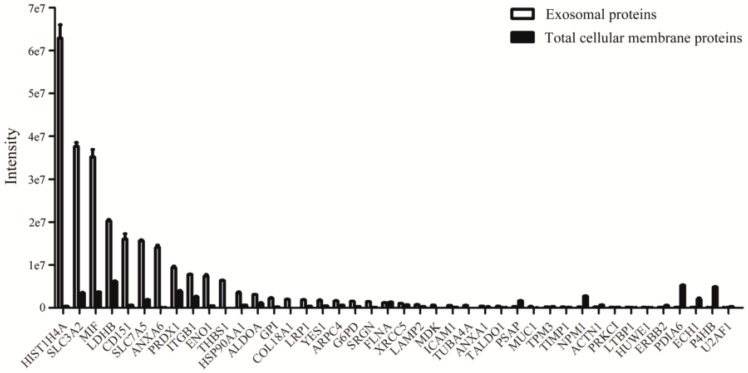
Proteins identified in the exosomes of NCI-H838 cells which were reported as potential biomarkers for lung cancer. Exosomes were prepared from supernatant of cell culture by ultracentrifugation method as described in the text. Trypsin-digested peptides were analyzed by data-independent analysis using Spectronaut software.

**Figure 3 ijms-20-00323-f003:**
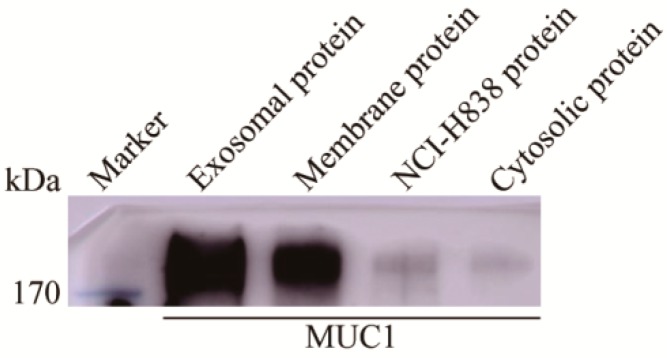
MUC1 was enriched in the exosomes of NCI-H838 cell line. Western blotting analysis was performed to measure expression of MUC1 in exosome, membrane, cytosolic and whole cell protein of NCI-H838 cells. 1 μg of each type of protein was separated by SDS-PAGE and 16A [41] monoclonal antibody was used as primary antibody for western blot.

**Figure 4 ijms-20-00323-f004:**
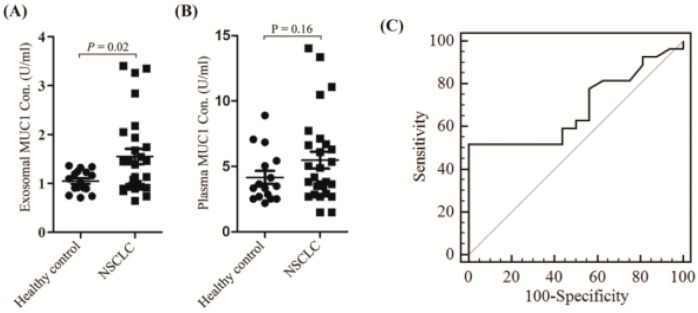
MUC1 in enriched exosome fractions and plasma of NSCLC patients and healthy controls. (**A**) Exosomal MUC1 levels in NSCLC patients and healthy controls; (**B**) Plasma MUC1 levels in the NSCLC patients and healthy controls; and, (**C**) receiver operating characteristic (ROC) curves based on exosomal MUC1 levels to differentiate NSCLC patients (*n* = 27) from healthy individuals (*n* = 16). The area under the curve was 0.685 (95% CI: 0.526–0.818, *p* = 0.0234). The different MUC1 levels of enriched exosome fractions were compared using unpaired student’s *t*-test between NSCLC and healthy control samples. A two-sided *p* value < 0.05 was defined as statistical significance.

**Table 1 ijms-20-00323-t001:** Proteins identified in the exosomes of NCI-H838 cells which were reported as potential biomarkers for lung cancer.

Protein Accessions	Protein Descriptions	Ratio of Mass Spectrometry Signal Intensity in Exosomal to Membrane Proteins	References
P48509	CD151 antigen (CD151)	27.092	[23]
P07996	Thrombospondin-1 (THBS1)	259.860	[24,25,26,27]
P06744	Glucose-6-phosphate isomerase (GPI)	10.700	[24]
P37837	Transaldolase (TALDO1)	8.424	[24]
Q5NKV8	Intercellular adhesion molecule 1 (ICAM1)	6.001	[24]
Q5U077	L-lactate dehydrogenase (LDHB)	3.307	[24]
Q60FE6	Filamin A (FLNA)	0.900	[24]
P12814	Alpha-actinin-1 (ACTN1)	0.251	[24]
P11413	Glucose-6-phosphate 1-dehydrogenase (G6PD)	5.411	[24]
Q14766	Latent-transforming growth factor beta-binding protein 1 (LTBP1)	2.894	[25]
Q59EN5	Prosaposin variant (PSAP)	0.176	[25]
Q01081	Splicing factor U2AF 35 kDa subunit (U2AF1)	0.182	[25]
Q5HYB6	Epididymis luminal protein 189 (TPM3)	0.838	[26]
Q7Z6Z7	E3 ubiquitin-protein ligase HUWE1 (HUWE1)	1.730	[26]
P10124	Serglycin (SRGN)	18.260	[26]
P14174	Macrophage migration inhibitory factor (MIF)	9.645	[27]
X5DNK3	Receptor protein-tyrosine kinase (Fragment) (ERBB2)	0.179	[27]
P62805	Histone H4 (HIST1H4A)	186.179	[28]
P39060	Collagen alpha-1(XVIII) chain (COL18A1)	133.480	[28]
P08133	Annexin A6 (ANXA6)	242.502	[28]
P68366	Tubulin alpha-4A chain (TUBA4A)	13.347	[28]
P59998	Actin-related protein 2/3 complex subunit 4 (ARPC4)	2.852	[28]
Q15084	Protein disulfide-isomerase A6 (PDIA6)	0.016	[28]
P07237	Protein disulfide-isomerase (P4HB)	0.011	[28]
Q6FGX5	TIMP metallopeptidase inhibitor 1, isoform CRA_a (TIMP1)	2.671	[29]
P21741	Midkine (MDK)	34.132	[29]
P06733	Alpha-enolase (ENO1)	16.102	[24,30]
P08195	4F2 cell-surface antigen heavy chain (SLC3A2)	10.947	[31]
P05556	Integrin beta-1 (ITGB1)	3.037	[31]
P13473	Lysosome-associated membrane glycoprotein 2 (LAMP2)	2.789	[31]
Q01650	Large neutral amino acids transporter small subunit 1 (SLC7A5)	8.272	[32]
P15941	Mucin-1 (MUC1)	8.977	[32,33,34]
P07900	Heat shock protein HSP 90-alpha (HSP90AA1)	5.526	[35]
P07947	Tyrosine-protein kinase Yes (YES1)	4.321	[35]
P41743	Protein kinase C iota type (PRKCI)	1.545	[35]
Q07954	Prolow-density lipoprotein receptor-related protein 1 (LRP1)	5.011	[36]
V9HWN7	Fructose-bisphosphate aldolase (ALDOA)	3.090	[24,36]
Q06830	Peroxiredoxin-1 (PRDX1)	2.424	[36]
Q5TZZ9	Annexin (ANXA1)	2.113	[37]
P13010	X-ray repair cross-complementing protein 5 (XRCC5)	1.456	[38]
P06748	Nucleophosmin (NPM1)	0.063	[39]
Q13011	Delta(3,5)-Delta(2,4)-dienoyl-CoA isomerase, mitochondrial (ECH1)	0.031	[40]

**Table 2 ijms-20-00323-t002:** Exosomal and plasma MUC1 in NSCLC patients and healthy controls.

	Number	Exosomal MUC1 Con. (U/mL)	Plasma MUC1 Con. (U/mL)	Exosome-Plasma MUC1 Ratio (%)
NSCLC patients	1	1.49	6.11	24.38
	2	2.05	6.63	30.97
	3	1.13	2.85	39.76
	4	0.94	4.90	19.10
	5	1.13	3.50	32.44
	6	1.94	7.14	27.15
	7	1.00	1.49	66.63
	8	0.74	1.49	49.43
	9	3.27	5.05	64.74
	10	3.35	14.06	23.86
	11	3.41	10.49	32.47
	12	0.94	2.69	34.77
	13	0.95	4.18	22.80
	14	1.54	11.09	13.89
	15	1.39	13.38	10.38
	16	0.89	3.50	25.53
	17	1.47	2.93	49.95
	18	0.85	2.69	31.55
	19	0.91	2.69	33.99
	20	1.45	6.31	23.00
	21	2.85	3.81	74.65
	22	1.74	7.73	22.51
	23	2.18	5.35	40.77
	24	0.64	3.66	17.64
	25	1.68	6.70	25.05
	26	0.89	3.81	23.42
	27	1.08	3.66	29.59
Healthy controls	1	1.19	5.43	21.93
	2	0.90	3.34	27.09
	3	0.91	2.52	36.17
	4	1.16	2.69	43.23
	5	1.34	4.13	32.41
	6	1.23	7.07	17.39
	7	1.36	8.90	15.29
	8	1.24	5.05	24.65
	9	1.09	3.34	32.79
	10	0.74	2.85	25.88
	11	1.32	6.85	19.31
	12	1.01	3.50	28.91
	13	0.89	2.52	35.37
	14	0.71	3.66	19.42
	15	0.75	2.19	34.27
	16	0.91	2.52	36.21

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
