# Peer review of "Preferential Localization of MUC1 Glycoprotein in Exosomes Secreted by Non-Small Cell Lung Carcinoma Cells"

_ijms, 2019, doi:10.3390/ijms20020323_

Round 1

Reviewer 1 Report

The manuscript is based on the premise that MUC1 is preferentially secreted in exosomes from NSCLC relative to cell surface expression.  More importantly in Figure 4 of their manuscript, they demonstrate that MUC1 expression on exosomes from NSCLC is higher than on exosomes from healthy control individuals, suggesting that exosomal MUC1 could be exploited as a biomarker for NSCLC.  ROC curves based on exosomal MUC1 levels to differentiate NSCLC patients from healthy individuals, gave area under the curve of 0.685.  Whereas this was not particularly impressive, the data suggested that exosomal MUC1 levels could serve as a biomarker for the disease.  The data as reported is not particularly innovative since others had previously reported MUC1 as a potential biomarker for NSCLC.  Nevertheless it is useful information which will help in describing more sensitive biomarkers for NSCLC which will allow the disease to be detected at early stages.  I have the following concerns that the authors should address.

1)     What is the rationale for choosing MUC1 as the exosomal marker that is preferentially localized in exosomes when other laboratories (such as Vykoukal et al.  Oncotarget 8, pg 95466-95480, 2017), have found it difficult to identify it in their exosomal preparations. 

2)     The 1.5 fold higher levels of MUC1 in exosomes from NSCLC patients relative to levels in exosomes from normal control subjects even though statistical significant given their sample size, does not appear to be particularly discriminative.  Alternatively do they just want to report that they are the first group to identify MUC1 in exosomes?  Sandfeld-Paulsen et al. (Molecular Oncology 10, 1595-1602, 2016), also identified MUC1 in exosomes from NSCLC.  So what is so original about their work.  This particular paper was not cited.  It should be cited and discussed relative to their work.

3)     In their Abstract, the sentence in line 19….Subsequent analysis of plasma exosomes…., appears to be at odds with the next sentence (line 22)…In contrast, no significant differences of MUC1…

4)     In Table 1, one of the columns is headed Ratio.  Ration of what??

5)     Figure 3 should be moved to supplemental data (seems out of place) because it adds little value to the manuscript as a whole.

6)     Glycan structures of MUC1 that they intend to study in future would have been more informative.  It would be interesting to see whether the glycoforms of MUC1 in exosomes from NSCLC are also different from the forms expressed in exosomes from normal control individuals?

Other minor concerns:

1)     Throughout the manuscript, there were many words that were not separated such as line 32 in the Results section should read localized in and not localizedin.

2)     In line 42 under Conclusions, …This is the first research… should be changed to …This is the first report.

Author Response

Response to Reviewer 1 Comments

Reviewer 1: The manuscript is based on the premise that MUC1 is preferentially secreted in exosomes from NSCLC relative to cell surface expression.  More importantly in Figure 4 of their manuscript, they demonstrate that MUC1 expression on exosomes from NSCLC is higher than on exosomes from healthy control individuals, suggesting that exosomal MUC1 could be exploited as a biomarker for NSCLC.  ROC curves based on exosomal MUC1 levels to differentiate NSCLC patients from healthy individuals, gave area under the curve of 0.685.  Whereas this was not particularly impressive, the data suggested that exosomal MUC1 levels could serve as a biomarker for the disease.  The data as reported is not particularly innovative since others had previously reported MUC1 as a potential biomarker for NSCLC.  Nevertheless it is useful information which will help in describing more sensitive biomarkers for NSCLC which will allow the disease to be detected at early stages.  I have the following concerns that the authors should address.

Point 1: What is the rationale for choosing MUC1 as the exosomal marker that is preferentially localized in exosomes when other laboratories (such as Vykoukal et al.  Oncotarget 8, pg 95466-95480, 2017), have found it difficult to identify it in their exosomal preparations.

Response 1: We thank reviewer for pointing out the discrepancy between our data and Vykoukal’s data. MUC1 is difficult to detect by mass spectrometry, so we propose to measure by antibodies. Technically, we used LTQ Orbitrap Fusion Lumos, which identified 3525 exosomal proteins from lung cancer cells. While Vykoukal et al. used LTQ-Orbitrap XL mass spectrometer, which identified 680 exosomal proteins from plasma of lung cancer patients.

Point 2: The 1.5 fold higher levels of MUC1 in exosomes from NSCLC patients relative to levels in exosomes from normal control subjects even though statistical significant given their sample size, does not appear to be particularly discriminative.  Alternatively do they just want to report that they are the first group to identify MUC1 in exosomes?  Sandfeld-Paulsen et al. (Molecular Oncology 10, 1595-1602, 2016), also identified MUC1 in exosomes from NSCLC.  So what is so original about their work. This particular paper was not cited.  It should be cited and discussed relative to their work.

Response 2: We apologize for the oversight. We have cited the paper by Sandfeld-Paulsen et al. and discussed the relevance of their findings to ours. The information of this particular paper is added into our revised Discussion as follows “Sandfeld-Paulsen et al. used 49 antibodies microarray, including an anti-MUC1 antibody, to capture MUC1 on the exosomes of NSCLC patients’ plasma, and the MUC1 positive detection rate is up to 80% [43]. This indicates that plasma exosomal MUC1 has potential as a diagnostic marker for NSCLC.

43.   Sandfeld-Paulsen, B.; Aggerholm-Pedersen, N.; Baek, R.; Jakobsen, K. R.; Meldgaard, P.; Folkersen, B. H.; Rasmussen, T. R.; Varming, K.; Jorgensen, M. M.; Sorensen, B. S. Exosomal proteins as prognostic biomarkers in non-small cell lung cancer. Mol. Oncol. 2016, 10, 1595-1602.

Point 3: In their Abstract, the sentence in line 19….Subsequent analysis of plasma exosomes…., appears to be at odds with the next sentence (line 22)…In contrast, no significant differences of MUC1…

Response 3: We thank reviewer for the correction. The sentence was revised to “In contrast, no significant difference of MUC1 level was found between NSCLC patients and healthy individuals’ plasma (mean value 5.48 ± 0.65 versus mean value 4.16 ± 0.49).

Point 4: In Table 1, one of the columns is headed Ratio.  Ration of what??

Response 4: We thank reviewer for the correction. The “Ratio” was changed to “Ratio of mass spectrometry signal intensity in exosomal to membrane proteins”.

Point 5: Figure 3 should be moved to supplemental data (seems out of place) because it adds little value to the manuscript as a whole. 

Response 5: We thank reviewer for this suggestion. Figure 3 was moved to supplemental data.

Point 6: Glycan structures of MUC1 that they intend to study in future would have been more informative.  It would be interesting to see whether the glycoforms of MUC1 in exosomes from NSCLC are also different from the forms expressed in exosomes from normal control individuals?

Response 6: We thank reviewer for this suggestion. It is very important to study glycan structures of exosomal MUC1 in physiological conditions. Currently, we are establishing methods to cleave glycans from exosomal MUC1 beginning from cancer cell lines. For exosomal MUC1 from plasma of normal control individuals, it is difficult to identify a normal cell type as control since MUC1-secreting exosomes are produced by many normal cell types.

Point 7: Throughout the manuscript, there were many words that were not separated such as line 32 in the Results section should read localized in and not localizedin. 

Response 7: We have corrected these errors.

Point 8: In line 42 under Conclusions, …This is the first research… should be changed to …This is the first report.

Response 8: The word has been changed accordingly.

Reviewer 2 Report

In this manuscript, Pan and colleagues illustrate that MUC1 is highly enriched in the exosomes isolated from a NSCLC cell line. They further demonstrate that exosomal MUC1 level is significantly different between NSCLC patients and the healthy controls. In general, the concepts of this manuscript is novel and interesting. However, I have several concerns on this manuscript.

1. Regarding the MUC1 - the authors need to provide evidence that validates the MS analysis result showing MUC1 is enriching in the exosomes. A WB experiment comparing MUC1 levels in exosomal, membrane and whole cell protein lysate. 

2. In the manuscript, as the authors focus many efforts on the exosmes, therefore, the authors need to describe in detail for example how many cells are seeded in what type of culture plate and what's the confluency, the methods for blood collection and plasma preparation.

3. When using ExoEasy or othe PEG based isolation methods, the authors need to keep in mind that these kits co-precipitate non-exosomal contaminants and can significantly interfere with your downstream analysis.

Author Response

Response to Reviewer 2 Comments

Reviewer 2: In this manuscript, Pan and colleagues illustrate that MUC1 is highly enriched in the exosomes isolated from a NSCLC cell line. They further demonstrate that exosomal MUC1 level is significantly different between NSCLC patients and the healthy controls. In general, the concept of this manuscript is novel and interesting. However, I have several concerns on this manuscript.

Point 1: Regarding the MUC1 - the authors need to provide evidence that validates the MS analysis result showing MUC1 is enriching in the exosomes. A WB experiment comparing MUC1 levels in exosomal, membrane and whole cell protein lysate. 

Response 1: We thank reviewer for this suggestion. The information of WB is added into our revised Results 2.2 as follows “The presence of MUC1 in the exosomes of NCI-H838 cells was further confirmed by western blot (Figure 3). Furthermore, the presence of MUC1 in the exosomes of NCI-H838 cells was confirmed by mapping for trypsin-digested peptide fragments (Figure S1).

Figure 3. MUC1 was enriched in the exosomes of NCI-H838 cell line. Western blotting analysis was performed to measure expression of MUC1 in exosome, membrane, cytosolic and whole cell protein of NCI-H838 cells. 1 µg of each type of protein was separated by SDS-PAGE and 16A monoclonal antibody was used as primary antibody for western blot.

Point 2: In the manuscript, as the authors focus many efforts on the exosmes, therefore, the authors need to describe in detail for example how many cells are seeded in what type of culture plate and what's the confluency, the methods for blood collection and plasma preparation. 

Response 2: We thank reviewer for this suggestion. The information of detail methods is revised accordingly in our revised Materials and Methods 4.4 as follows “NCI-H838 cells were grown in 10 cm dishes with serum-free RPMI-1640 media after 72 hours, and the cells reached a confluency of 90-100%. Then the media was collected and centrifuged at 3000 g at 4 °C for 10 min to remove detached cells. Green-top tube (containing sodium heparin) as an anticoagulant plasma separator tubes were used to collect blood samples. The blood samples were then centrifuged at 5000 rpm for 10 min to collect the plasma, which was stored at -80 °C until used.

Point 3: When using ExoEasy or other PEG based isolation methods, the authors need to keep in mind that these kits co-precipitate non-exosomal contaminants and can significantly interfere with your downstream analysis.

Response 3: Thanks for the reviewer’s comment. Compared to ultracentrifugation method, ExoEasy Maxi Kit isolation method may cause some non-exosomal contaminants. In order to minimize the impact of downstream analysis, we washed the spin column by 10 ml buffer XWP. We also highlighted this in the text and Figure 4 legend that the ExoEasy purification method is a method which enriches exosomes. 

Round 2

Reviewer 2 Report

This manuscript can be accepted after revision.